# A novel hypervariable variable number tandem repeat in the dopamine transporter gene (*SLC6A3*)

Abner T Apsley[1,2] , Emma R Domico[1] , Max A Verbiest[3,4,5] , Carly A Brogan[1], Evan R Buck[1] , Andrew J Burich[6], Kathleen M Cardone[1], Wesley J Stone[1], Maria Anisimova[3,5] , David J Vandenbergh[1,2,7,8]

The dopamine transporter gene, *SLC6A3*, has received substantial attention in genetic association studies of various phenotypes. Although some variable number tandem repeats (VNTRs) present in *SLC6A3* have been tested in genetic association studies, results have not been consistent. VNTRs in *SLC6A3* that have not been examined genetically were characterized. The Tandem Repeat Annotation Library was used to characterize the VNTRs of 64 unrelated long-read haplotype-phased *SLC6A3* sequences. Sequence similarity of each repeat unit of the five VNTRs is reported, along with the correlations of SNP–SNP, SNP–VNTR, and VNTR–VNTR alleles across the gene. One of these VNTRs is a novel hyper-VNTR (hyVNTR) in intron 8 of *SLC6A3*, which contains a range of 3.4–133.4 repeat copies and has a consensus sequence length of 38 bp, with 82% G+C content. The 38-base repeat was predicted to form G-quadruplexes in silico and was confirmed by circular dichroism spectroscopy. In addition, this hyVNTR contains multiple putative binding sites for PRDM9, which, in combination with low levels of linkage disequilibrium around the hyVNTR, suggests it might be a recombination hotspot.

## Introduction

Variable number of tandem repeat loci (VNTR) are important sites of genomic variation (Gall-Duncan et al, 2022; Xiao et al, 2022). VNTRs are defined as regions of DNA where a particular nucleotide sequence is repeated in tandem and the number of copies of the repeated sequence varies between individuals (Hannan, 2018). Previous studies have shown that VNTRs play a role in various biological processes such as the formation of G-quadruplexes (G4s) (Li et al, 2016; Guiblet et al, 2021), DNA I-motifs (Kondo et al, 2004),

recombination hotspots (Zavodna et al, 2018), gene expression control (Lalioti et al, 1997; Borel et al, 2012; Li et al, 2016; Örd et al, 2020; Johansson et al, 2022), alteration of DNA methylation (Garg et al, 2021), and histone modifications (Vasiliou et al, 2012). Interpersonal variation in VNTRs can have functional consequences by altering gene expression of nearby genes as is seen for three VNTRs in and around the arginine vasopressin receptor 1A (*AVPR1A*, or proposed nomenclature *VTR1A* [Theofanopoulou et al, 2021]), which have been associated with altered gene expression and the complex trait of externalizing behavior (Landefeld et al, 2018). In addition to playing a role in biological processes, VNTRs have been implicated in psychological disorders (Hannan, 2018, 2021; Gall-Duncan et al, 2022), such as Alzheimer's disease (De Roeck et al, 2018), schizophrenia (Song et al, 2018), amyotrophic lateral sclerosis (Course et al, 2020), and myopathy with rimmed ubiquitin-positive autophagic vacuolation (Ruggieri et al, 2020). These results indicate that it is crucial to genotype polymorphic tandem repeat loci accurately to get a full picture of the genetic component of phenotypic variation in the human population.

Many different characteristics, such as location, total length, consensus sequence motif, intrapersonal repeat copy number variation, and conservation or purity of consensus sequence motif, may contribute to how VNTRs influence phenotypes (Hannan, 2018). Because of the repetitive nature of VNTRs, genomic functional elements that require specific DNA motifs in tandem are likely candidates for mechanisms by which VNTRs play a role in disease/phenotype determination. For example, G4s are four-stranded secondary DNA structures that form in the presence of a repetitive guanine-rich DNA motif (Huppert & Balasubramanian, 2005) and have been shown to influence biological processes such as transcription rates (Huppert & Balasubramian, 2005; Brázda et al, 2019); therefore, specific VNTR consensus sequences may aid in the formation of G4s. In addition, the presence of multiple VNTR consensus sequences that contain protein-binding site motifs may

[1]Department of Biobehavioral Health, The Pennsylvania State University, State College, PA, USA  [2]The Molecular, Cellular and Integrative Biosciences Program, The Pennsylvania State University, State College, PA, USA  [3]Institute of Computational Life Science, School of Life Sciences and Facility Management, Zürich University of Applied Sciences, Wädenswil, Switzerland  [4]Department of Molecular Life Sciences, Faculty of Science, University of Zurich, Zurich, Switzerland  [5]Swiss Institute of Bioinformatics, Lausanne, Switzerland  [6]Department of Information Science and Technologies - Applied Data Sciences, The Pennsylvania State University, State College, PA, USA  [7]Institute of the Neurosciences, The Pennsylvania State University, State College, PA, USA  [8]The Bioinformatics and Genomics Program, The Pennsylvania State University, State College, PA, USA

Correspondence: djv4@psu.edu
Emma R Domico's present address is Armstrong Institute for Patient Safety and Quality, Johns Hopkins University, Baltimore, MD, USA

**Table 1. Consensus length, number of different alleles, copy number of the major allele, degree of heterozygosity, genic location, and chromosomal coordinates for each VNTR.**

| VNTR characteristics | | | | | | |
|---|---|---|---|---|---|---|
| Tandem repeat ID | Consensus length (bp) | Number of different alleles | Copy number of major allele | Degree of heterozygosity | Location | Hg38 Chr5 coordinates |
| TR09 | 66 | 2 | 7 | 0.469 | Intron 3 | 1435680–1436128 |
| TR17 | 38 | 13 | 28 | 0.781 | Intron 4 | 1422352–1423382 |
| TR21 | 38 | 43 | 3 | 0.938 | Intron 8 | 1414387–1414518 |
| TR22 | 30 | 4 | 6 | 0.531 | Intron 8 | 1411741–1411922 |
| TR30 | 40 | 3 | 10 | 0.406 | 3'UTR | 1393582–1393985 |

Degree of heterozygosity calculated as the fraction of individuals who were heterozygous for the given locus.

**Table 2. Table shows the consensus sequence and percent G+C for each VNTR.**

| VNTR consensus sequences | | |
|---|---|---|
| ID | Consensus sequence | GC% |
| TR09 | TGGCCACCACCGTTCAAGGGAGCCATTTCCTCACCCAGGTGCCCAGGGAAGCATCCAGGAGGGGAC | 63% |
| TR17 | TGTGGGCAGCGGTGGGTACCCAGCACCGTGGGCAGCAC | 70% |
| TR21 | CCCCCACCCAGCGCCTTCCCCGCCCTGCCCCTCCAGGC | 82% |
| TR22 | TGTGTCTGAGTGTGTATGTTGCATGGTATG | 42% |
| TR30 | AGGAGCGTGTACTACCCCAGGACGCATGCAGGGCCCCCAC | 67% |

Consensus sequences of the human reference genome (GRCh38/hg38) VNTRs are shown in the genomic orientation (antisense to *SLC6A3*).

provide a source of variation in their binding affinity (Vasiliou et al, 2012).

*SLC6A3*, which encodes the dopamine transporter protein (DAT1), has been studied extensively in relation to its VNTRs. *SLC6A3* has 15 exons and spans ~52.5 kb in length on human chromosome 5 (GRCh38/hg38: chr5:1,392,794–1,445,440). It contains a VNTR in the 3'-UTR of the gene (Vandenbergh et al, 1992) with repeat copy numbers ranging from 3 to 11 and a consensus sequence of 40 bp in length (see Tables 1 and 2, TR30). Genetic analyses suggested association of this polymorphic site with various phenotypes such as attention deficit hyperactivity disorder (Cook et al, 1995; Bidwell et al, 2011), Parkinson's disease (du Plessis et al, 2020), hypertension (Kim et al, 2017), depression (Kirchheiner et al, 2007), substance use disorders (van der Zwaluw et al, 2009), and other physiological and psychological ailments (Salatino-Oliveira et al, 2018). A second VNTR in intron 8 of *SLC6A3*, with repeat copy numbers of either 5 or 6 and a consensus repeat sequence length of 30 bp (see Tables 1 and 2, TR22), was associated with cocaine dependence in Brazilian individuals (Guindalini et al, 2006). This intron 8 VNTR was also tested in many other studies for its association with disease-related phenotypes (see Salatino-Oliveira et al, 2018 for an extensive review). A third VNTR located in intron 3 of *SLC6A3* was reported to have a consensus sequence length of 63 bp and repeat copy numbers of 7 and 8 (see Tables 1 and 2, TR09) (Franke et al, 2008). Finally, in 2017, Kim and colleagues reported on a fourth VNTR present in intron 4 of *SLC6A3*, which had a consensus sequence of 75 bp and repeat copy numbers ranging from 11 to 32 (see Tables 1 and 2, TR17) (Kim et al, 2017). Recent work by Course and colleagues has also documented the presence of this intron 4 VNTR using long-read haplotype-phased assemblies; however, they report it as having a 38-bp consensus sequence length, essentially dividing the

intron 4 VNTR reported by Kim and colleagues in half (see Tables 1 and 2, TR17) (Course et al, 2021).

Although *SLC6A3* has received a large amount of attention in relation to its VNTR landscape and the associations of these VNTR alleles with phenotypes of interest, there have been both inconsistent and contradicting results reported. For example, in meta-analyses and systematic reviews, alcohol addiction has been reported to be both significantly and nonsignificantly associated with alleles at the 3'-UTR VNTR site (van der Zwaluw et al, 2009; Ma et al, 2016). One possible explanation for these inconsistent and contradictory reports may be that there is a lack of complete characterization of genetic variation in this gene. In fact, in 1993, Byerley and colleagues, using Southern blot techniques with TaqI-digested DNA, reported a site in the 3'-half of *SLC6A3* with alleles that varied by as much as 7.1 kb (Byerley et al, 1993) but was not studied further. We hypothesized the presence of one or more VNTRs in this region of *SLC6A3* that may account for the heterogeneity in sequence length reported by Byerley et al, but that this VNTR had escaped detection due to the lack of technology sufficient to fully characterize its total repeat length.

In the present work, we used 64 publicly available long-read haplotype-phased genome assemblies from 32 individuals from the Human Genome Structural Variation Consortium (Ebert et al, 2021) and the Tandem Repeat Annotation Library (TRAL) Python library (Delucchi et al, 2021) to characterize the VNTR landscape of *SLC6A3*. We report the discovery of a novel hypervariable number tandem repeat (hyVNTR) in the intron 8 region of *SLC6A3* that has a consensus sequence length of 38 bp and repeat copy numbers ranging from 3.4 to 133.4. In addition, we demonstrate its ability to form G-quadruplexes in vitro. We also report linkage disequilibrium (LD) values of each *SLC6A3* VNTR with all other SNPs and VNTRs present

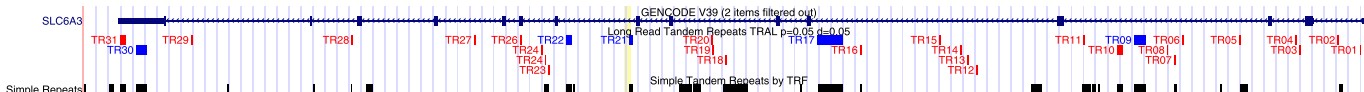

**Figure 1.   UCSC Genome Browser view of SLC6A3 with UCSC's Simple Repeats track (TRF annotation of sequence) and a custom track showing the TRs detected by the TRAL with non-variable TRs in red and VNTRs in blue.**

in the gene, with emphasis on the hyVNTR. Potential PRDM9-binding sites—a known site for initiation of recombination—are present in this hyVNTR. These data suggest the need to assess potential functional relevance of the intron 8 hyVNTR and its role in the genetic underpinnings of dopamine-related traits.

# Results

### *SLC6A3* has a highly variable sequence length

The tandem repeat architecture of *SLC6A3* was analyzed using long-read haplotype-phased genome assemblies from 32 unrelated individuals (Ebert et al, 2021). The assemblies were created by either CLR (continuous long reads) or CCS (circular consensus sequencing) technologies (Pacific Biosciences). Some assemblies were available for both technologies, and comparison of the two versions revealed an average similarity score of 99.96% (SD = 0.06%). Using the length of *SLC6A3* in the human reference genome (GRCh38/hg38), a similarity score at this value corresponds to an average of 22.67 (SD = 29.06) mismatching bases per *SLC6A3* sequence. In addition, an average of 20.94 (SD = 19.69) indels were observed across all *SLC6A3* sequences. Individual haplotype comparison values are shown in Table S1.

No CNVs or large deletions were found where *SLC6A3* is located, and all haplotype-phased long-read genome assemblies contained one copy of the gene. *SLC6A3* sequences ranged from 52,673 to 58,846 bp (mean = 54,749 bp, SD = 1,265 bp), confirming the presence of heterogeneity in sequence length among our samples.

### Tandem repeat annotations reveal a novel hypervariable VNTR located in intron 8 of *SLC6A3*

Many algorithms have been used to detect tandem repeats (TRs) in biological sequences, leading to inconsistent TR annotations of identical sequences (Schaper et al, 2012). To address this issue, we used the TRAL Python library (Delucchi et al, 2021), which compares, harmonizes, and makes nonredundant, the output of different repeat detection algorithms (TRF, XSTREAM, PHOBOS, and T-REKS; see the Materials and Methods section). We used TRAL to annotate all *SLC6A3* sequences for repeats. TRs were distributed across the length of the gene with no obvious pattern. Fig 1 shows the location of each invariant TR (red) and VNTR (blue) displayed above the annotations from the standard tandem repeat finder (TRF) results. Table 1 shows exon/intron locations, repeat parameters, and allele information for each TR that was annotated as a VNTR. The same information for TRs that had the same copy number across all *SLC6A3* sequences is included in Table S2. Most of the TRs had a consensus sequence length less than or equal to 10 bp, with 8 TRs

having more than 10 bp in their consensus sequence. Only 5 TRs contained more than one copy number allele (TRs 09, 17, 21, 22, and 30) and were, therefore, designated as VNTRs (see Fig 1). Other than TR26, which is located in exon 10, all TRs were located either in introns or the 3′UTR of the gene. Table 2 shows the consensus sequence for each annotated VNTR.

Given that four of the VNTRs (TR09, TR17, TR22, and TR30) have been characterized through PCR amplification (Vandenbergh et al, 1992; Guindalini et al, 2006; Kim et al, 2017), we focused on a novel hyVNTR located in intron 8 (TR21), upstream of the published VNTR in this intron (Guindalini et al, 2006). This site was not amplified by PCR despite numerous attempts using more than seven different commercial DNA polymerase kits (data not shown). We predict that the polymerases tested were blocked from completing synthesis of the tandem repeats by their high G+C concentration (82%) and their potential to form G-quadruplexes in the presence of potassium (see below and Fig 6), which is found in many PCR buffers. The hyVNTR in intron 8 had alleles ranging from 3.4 to 133.4 copies of the consensus sequence in the long-read sequence data. The human reference genome (GRCh38/hg38) shows this repeat as having 3.4 copies, which was the smallest and most common (n = 4, 6.1%) version of the repeat detected.

### Repeat copy number and repeat sequence purity are highly variable across *SLC6A3* VNTRs

To compare both the individual repeat sequences within each VNTR, and these sequences across all *SLC6A3* assemblies, we generated what we have termed "Mola" charts, which graphically demonstrate alignment similarity between each individual repeat sequence unit of the VNTRs and their associated consensus sequence (see the Materials and Methods section for details). We generated two general types of Mola charts, a multi-color chart that displays a random color for each unique repeat sequence across our sample, and a 3-color chart, with a color gradient of blue showing a high sequence similarity and red showing a weak sequence similarity to the repeat's consensus sequence. A complete documentation of each VNTR's Mola charts can be found in Table S3A–C.

Multi-color Mola charts for each VNTR are shown in Fig 2. Two of the VNTRs displayed both a large number (206 for TR21 [Fig 2A] and 91 for TR17 [Fig 2B]) and a large proportion of unique repeat sequences (14% for TR17 and 71% for TR21). Only one individual was homozygous for TR21 based on copy number, and this person's alleles differed based on the sequence of the units within the alleles. In contrast, most of the VNTRs had relatively few differences in repeat copy number (length differences) and fewer unique sequences (color differences) (Fig 2C–E). To determine whether the variation within the 38 bp units of TR21 was located within

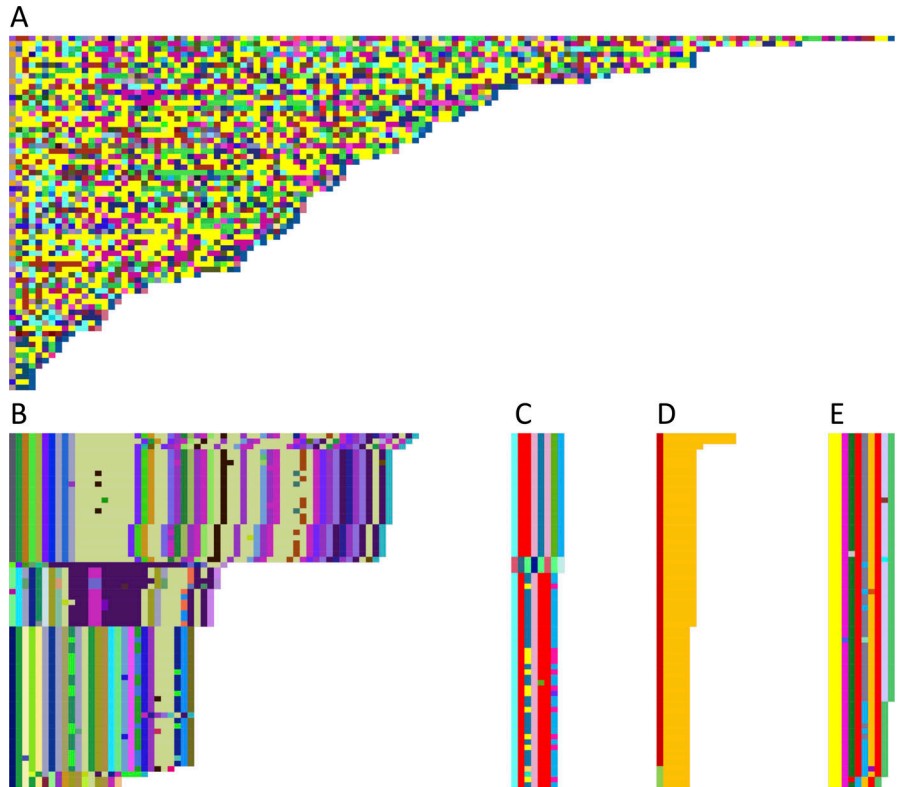

**Figure 2. Multi-color Mola charts illustrating the degree of variability of each repeat unit in the 5 VNTRs within SLC6A3.**
**(A)** TR21, **(B)** TR17, **(C)** TR09, **(D)** TR22, and **(E)** TR30. Full names of haplotypes (each row) are available in Table S3B. Haplotypes are sorted based on the number of repeat units. The length of each unit varies because of insertions or deletions.

homopolymer runs, we performed two analyses. First, a sensitivity analysis was conducted by excluding homopolymer runs in repeat sequences of TR21. When excluding homopolymer runs of four nucleotides or longer, only six of the unique sequences were the same as others, leaving 200 unique TR21 repeat sequences. This minimal change in the number of unique repeat sequences, after excluding homopolymer runs, suggests that the variation observed in TR21 was not due to sequencing errors. Second, we compared CLR and CCS sequencing results from TR21 in 12 sequences. These 12 sequences showed a variation in the presence of discrepancies. The maximum number of mismatches was 2 bp, with most comparisons having no mismatches. The maximum number of indels was 39, with 34 of the mismatches being in a homopolymer stretch. The mean number of indels was 11.9 and most of these indels were in homopolymer runs—typically homopolymer stretches of C followed by an A; however, they are few in number and not likely to have a large influence on the number of unique sequences in TR21. Additional analysis of TR21 in parent–child pairs generated similar results (see details below in *SLC6A3* VNTRs, which show stable copy number inheritance patterns section).

Three-color Mola charts for each VNTR are shown in Fig 3. The colors are based on sequence similarity to the individual VNTR's consensus (global alignment score, EMBOSS needle, Rice et al, 2000), which emphasizes the relative degree of variation from the consensus present in each VNTR. The greatest differences in repeat sequences from the consensus sequence, for each VNTR, are as follows: TR09 gaps/mismatches = 26, TR17 gaps/mismatches = 27, TR21 gaps/mismatches = 15, TR22 gaps/mismatches = 3, and TR30

gaps/mismatches = 5. In the case of TR21, no clustering of haplotypes is apparent (Fig 3A); however, even for the highly variable TR17 (Fig 3B) there are blocks of similarity both in length of the repeat and in similarity of the units. Similarities in both length and number of units can also be seen for TR09, TR22, and TR30 (Fig 3C–E).

In addition to constructing multi-color and three-color Mola charts, we also tested whether alleles to each VNTR could be grouped based on ancestry or 1000 Genomes Project superpopulation. Ancestry groups included African (haplotype sample size n = 16), South Asian (n = 10), East Asian (n = 12), European (n = 12), African admixed (n = 2), and Indigenous admixed (n = 10). After excluding the African admixed ancestry category because of a low sample size, the ancestry group did not predict TR21 copy number ($F(4,55) = 1.77$, $P = 0.15$). Results for TR21 alleles grouped by ancestry can be found in Table S4A and B.

### *SLC6A3* VNTRs show stable copy number inheritance patterns

Included with the 64 unrelated *SLC6A3* sequences were three parent–child trios (HG00512, HG00513, and HG00514; HG00731, HG00732, and HG0073; and NA19238, NA19239, and NA19240). All VNTR copy number alleles were inherited stably from parent to offspring. Global alignments of each child's *SLC6A3* sequence with the corresponding sequence inherited from each parent show an average of 99.95% similarity. In all samples, TR09, TR17, TR22, and TR30 had no intergenerational mismatches, insertions, or deletions; however, TR21 displayed between 1 and 5 insertions or deletions between generations, suggesting genomic instability at this site (Table S5).

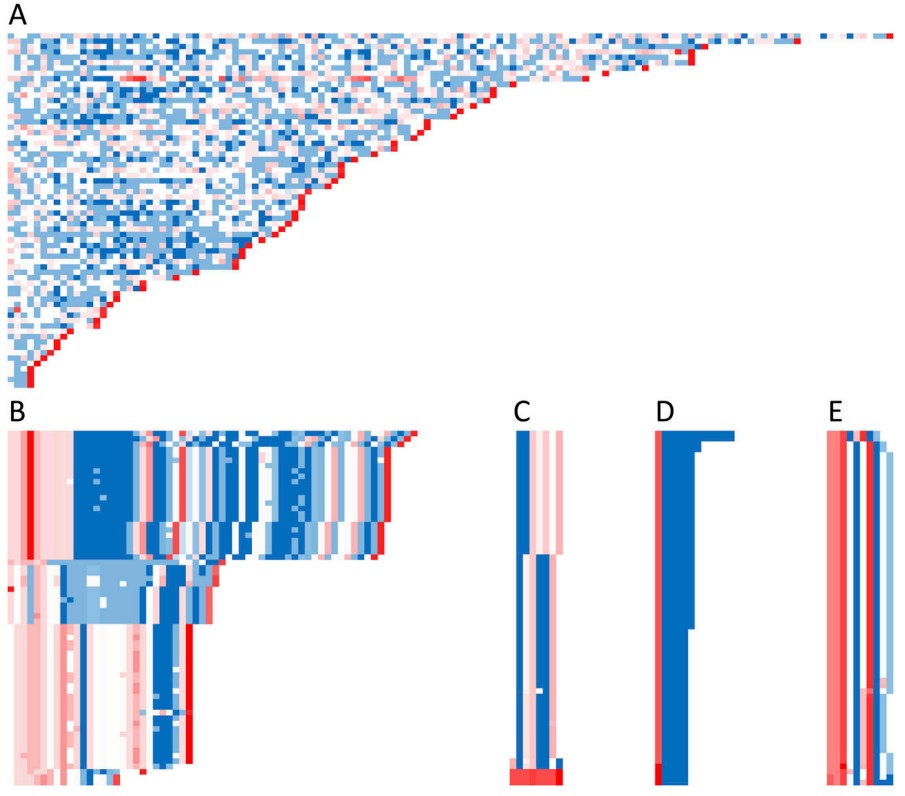

**Figure 3. Three-color Mola charts illustrating the modes of variability in the five VNTRs within SLC6A3.**
(A) TR21, (B) TR17, (C) TR09, (D) TR22, and (E) TR30. Details are as in Fig 2, but the haplotypes are sorted based on similarity of the first repeat units in a haplotype, except for TR21 (A) and TR17 (B), which are sorted by number of repeat units due to their complexity. (Full names of the sequences and their alignment scores for each repeat are available in Table S3C).

Additional analysis of the two types of sequencing methodologies was carried out with a parent's allele and their child's homologous allele (parent-child pairs). The CCS group was calculated to have an average of 1.8 ± 0.7 gaps in the parent–child pairs. The mean value for the number of gaps between parent–child pairs in the CLR group was 17 ± 5.2. Although only six total parent–child pairs were used in this comparison, there was a clear trend since the CLR group had more gaps than the CCS group in every pairing while also having nearly 10 times the total number of average gaps. There were also no mismatches seen in any of the CCS parings, whereas one of the parent–child pairs in the CLR group had 2 mismatches.

### *SLC6A3* hypervariable VNTR (TR21) is in a region of low linkage disequilibrium

Across the length of the gene, a total of 108 SNPs (excluding any SNPs found within annotated VNTRs) with a minor-allele frequency of greater than 5% in our sample were observed (see the Materials and Methods section for details on annotation of SNPs and determination of SNP allele frequencies). Ninety-three of the observed SNPs are present in dbSNP (Build 153, https://www.ncbi.nlm.nih.gov/snp/). Correlations among SNP–SNP, SNP–VNTR, and VNTR–VNTR pairs were calculated and are shown in Fig 4 (see Table S6 for more detailed information). TR09 (intron 3) and TR17 (intron 4) fell within the same haplotype block and had an $r^2$ value of 0.96. All other VNTR–VNTR pairs had $r^2$ values of less than 0.08. Two SNPs (rs458609 and rs393795) were highly correlated with both TR09 and TR17 (all $r^2 > 0.96$). In contrast, the highest SNP–VNTR correlations for TR21, TR22, and TR30 were

rs2937640-TR21: $r^2$ = 0.11, rs10074171-TR22: $r^2$ = 0.24, and rs11564775-TR30: $r^2$ = 0.49, respectively. All the remaining SNP–VNTR correlations for TR21, TR22, and TR30 were below 0.39. The two closest SNPs flanking TR21 (rs11564759 and rs59133686) showed correlation values with TR21 of 0.016 and 0.001, respectively; these two SNPs showed a correlation value of 0.003 with one another, despite being less than 800 nucleotides on either side of TR21. Because of the low correlations between SNPs and VNTRs in this region, we also report correlation values for all SNPs within introns 7 and 8 separately (see Table S7). The variation in LD in this region led to a search for binding sites for PRDM9 (PR/SET domain 9 protein), which plays a critical role in initiation of crossing over (Cheung et al, 2010; Myers et al, 2010; Altemose et al, 2017). The Genome Browser's JASPAR Transcription Factors track for hg38 reveals 139 sites across the entire gene, 6 of which are present in TR21, which only contains 3.4 copies of the repeat. The haplotype with the largest number of repeats (133.4) is predicted to contain 421 PRDM9-binding sites (Table S8) using MEME (Bailey & Elkan 1994). This enrichment of PRDM9 sites within TR21, coupled with the low levels of LD within the region suggest crossing over occurs frequently in and around this site.

### G-quadruplexes are present in *SLC6A3* novel hypervariable VNTR

To explore the functional relevance of the VNTRs we detected, we used the G4-Hunter web application (Brázda et al, 2019) to predict the formation of G4s in the human reference genome (GRCh38/hg38 with 3.4 copies of TR21) and alternate human reference genome (KI270791v1_alt: 64,333–121,212 with 75.4 copies of TR21). The human reference genome results showed no regions of *SLC6A* having over

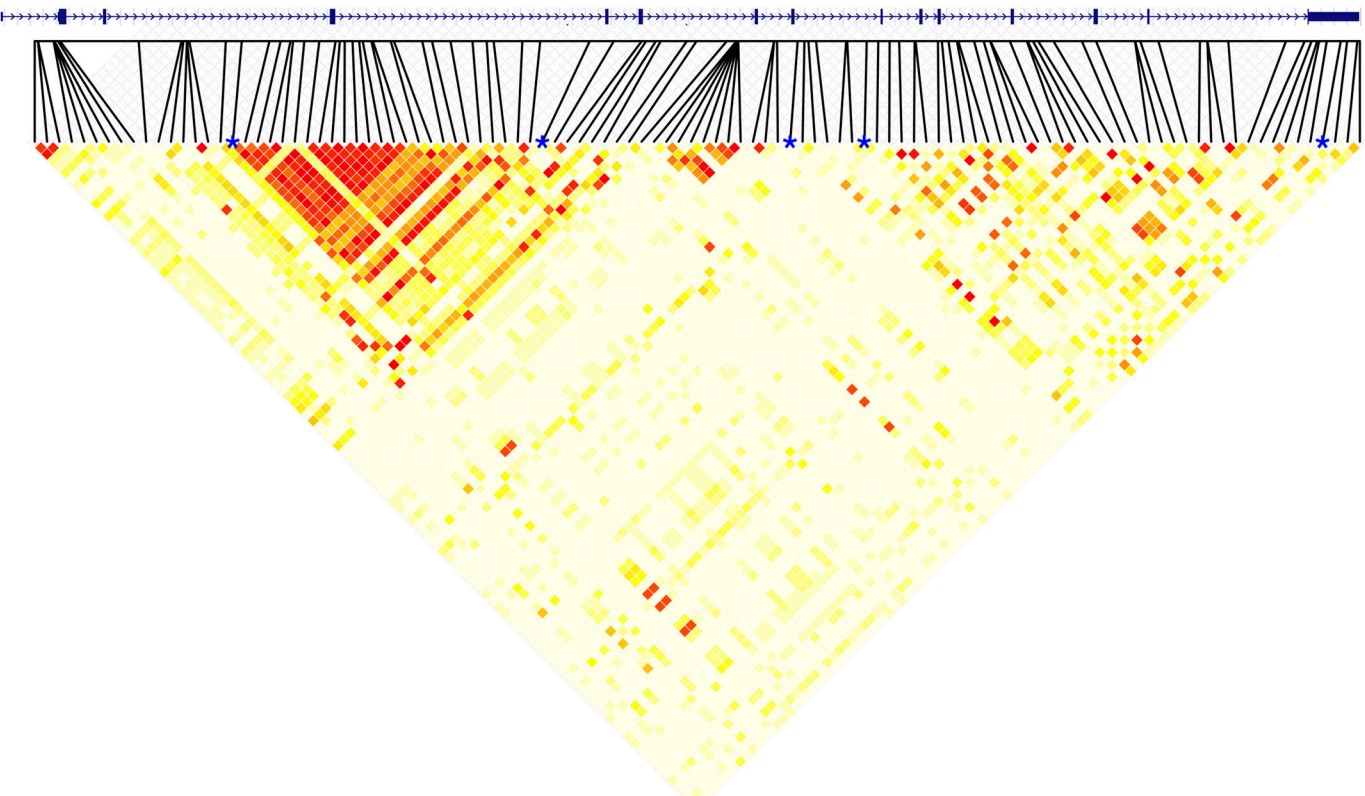

**Figure 4. Heatmap of correlations between all SNPs and VNTRs in SLC6A3.**
The gene is shown in the 5′ to 3′ orientation above the heatmap, with black lines indicating the location of each SNP. The five VNTRs are denoted with a blue asterisk above the heatmap, with TR09 on the far left and TR30 on the far right. On the heatmap, red indicates a stronger correlation and white indicates a weaker correlation.

50% coverage by predicted G4s (Fig 5A, yellow). In contrast, one distinct region in the alternate human reference genome *SLC6A3* sequence displayed greater than 50% coverage by G4s (Fig 5A, purple). The coverage peak (which spans ~3,000 bp) begins at gene nucleotide coordinate 33,417 (genome coordinates chr5: 1,412,023–1,415,023, GRCh38/hg38) and corresponds with the hyVNTR in intron 8 (TR21), with an average G4 coverage of 99%. A similar peak can be observed in Fig 5B, where the human reference genome is predicted to form around 6 G4s in TR21 at its highest, whereas the alternate human reference genome is predicted to form ~25.

In addition to using in silico methods to predict the presence of G4s, circular dichroism (CD) spectroscopy was used to confirm the presence of the G4s predicted in TR21. An oligonucleotide of 38 bases matching the G-rich strand of the TR21 consensus sequence generated characteristic spectra of G4 structures (i.e., 210 and 260 nm peaks and a 240 nm trough) (Kypr et al, 2009) (Fig 6). The signal strength at the two peaks and the trough was increased with addition of potassium-containing buffer and further with additional potassium chloride, consistent with the presence of one or more G4s.

## Discussion

We used the TRAL to detail the VNTR landscape of *SLC6A3*—a frequent candidate for genetic association studies—in 64 publicly

available long-read haplotype-phased genome assemblies. We found a total of 31 TRs present in more than 95% of our sample, with the exception of one TR (TR11), which was weakly conserved and identified in only ~50% of the sequences. Five of these TRs were variable in copy number. One of these 5 is a novel hyVNTR (TR21) with a consensus sequence of 38 bp, which presents new genetic variation within *SLC6A3* to be tested. This hyVNTR was highly variable in repeat copy number, displaying from 3.4 to 133.4 repeat copies in our sample, and had the highest degree of heterozygosity (Table 1) when compared with the four other VNTRs.

This VNTR also allowed an analysis of the CLR and CCS methods of long-read sequencing. We found that almost all of the gaps were seen in long homopolymer C stretches followed by an A. These homopolymer stretches are known to be difficult to sequence accurately, so it is reasonable that these gaps are misreads; however, these homopolymer stretches also have a high rate of polymorphism due to replication slippage causing actual misreads of the DNA by the replication machinery. In addition, the lower alignment seen in the CLR group in the parent–child pairs implies that it is not as accurate of a sequencing method as CCS. Although it is possible that the CCS sequencing is misregistering actual polymorphisms that CLR sequencing did not, we are unable to consider all gaps seen as just misreads, and we do not have the data to resolve this issue.

To the best of our knowledge, TR21 might represent a variant previously described from Southern blots (Byerley et al, 1993);

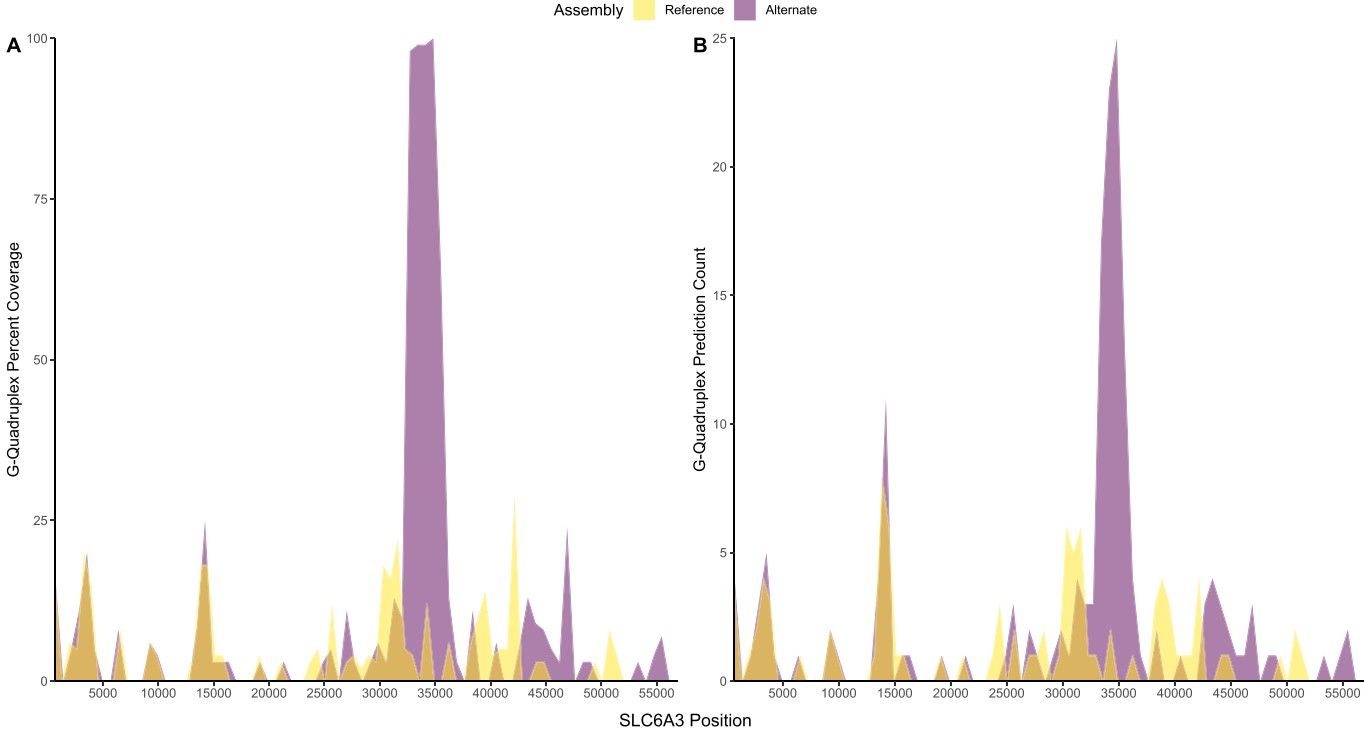

**Figure 5. G4-Hunter results for SLC6A3 sequences from the human reference genome (GRCh38/hg38; yellow) and the alternate human reference genome (KI270791v1; purple).**
**(A)** shows the G4 coverage percentage in each section of SLC6A3. **(B)** shows the G4 count in each section of SLC6A3.

however, there are two limitations that make the relation unclear. First, the cDNA probe used by Byerley and colleagues covered the last nine transmembrane domains (exons 4–13) and the introns (Donovan et al, 1995), in which there are three VNTRs (TR17, TR21, and TR22), all of which could contribute to allele sizes reported by Byerley. Second, the alleles in Byerley et al vary by 500 base pairs (possible limit of resolution of Southern blots), whereas the alleles of TR21 vary by 38 base pairs (this study). Thus, the alleles detected by Byerley might include multiple sites of variation. The longest and shortest alleles in our data differ by ~5 kb, whereas the alleles in Byerley et al vary by 7.1 kb, suggesting that longer alleles at TR17, TR21, and TR22, might exist.

Most studies of VNTRs to date have focused on their variation in repeat copy number; however, more recent data demonstrate that repeat sequence "purity" often varies within VNTRs, and this variation can be functionally relevant (Song et al, 2018; Course et al, 2020; Gall-Duncan et al, 2022). In addition to the variability in VNTR copy numbers observed, individual VNTR repeats in *SLC6A3* showed a large range in the amount of sequence heterogeneity present, with TR21 having the largest amount of heterogeneity (Fig 2). Although not tested in the present work, disruptions of PRDM9-binding site motifs in TR21 may take place as a result of the impurity of this hyVNTR's repeat sequences.

We found two general LD blocks in *SLC6A3*, the stronger of the two containing both TR09 and TR17. Two SNPs (rs458609 and rs393795) were also highly correlated with both TR09 and TR17; however, the three other VNTRs (TR21, TR22, and TR30) were not in

high LD with any SNPs. TR21 and TR22 were both found in a stretch of low LD spanning introns 7 and 8, whereas TR21, TR22, and TR30 were found in the weaker LD block. TR21, the novel hyVNTR, had the smallest maximum correlation with any SNP in the gene body, similar to the VNTR in *TCHH* (Mukamel et al, 2021). This finding suggests the necessity of targeted long-read sequencing of this region in future association studies because the variability in this location is not captured by nearby SNPs. In addition, previous work has documented the presence of recombination hotspots throughout the body of *SLC6A3*, specifically near the site of TR21 (Zhao et al, 2019). Our report on the low LD values of TR21 and its neighboring SNPs, along with the density of potential PRDM9-binding sites within the hyVNTR, provide additional evidence that TR21 might be the site of frequent recombination which might contribute to the heterogeneity of copy numbers found in this hyVNTR.

Although each human genome has interpersonal variation, not all variants are biologically or functionally relevant. In addition to searching for recombination-related factors, we also tested for the potential of TR21 to form G4s, which have been identified to play many distinct biological roles in genome function and regulation of gene expression (Li et al, 2016; Kwok & Merrick, 2017). We predicted and experimentally confirmed the presence of G4s—which could potentially alter transcription rates, mRNA splicing (Verma & Das, 2018), or both—in TR21. The high GC content of this hyVNTR may also explain why the region has not yet been studied using PCR techniques. Additional studies are needed to assess the relation

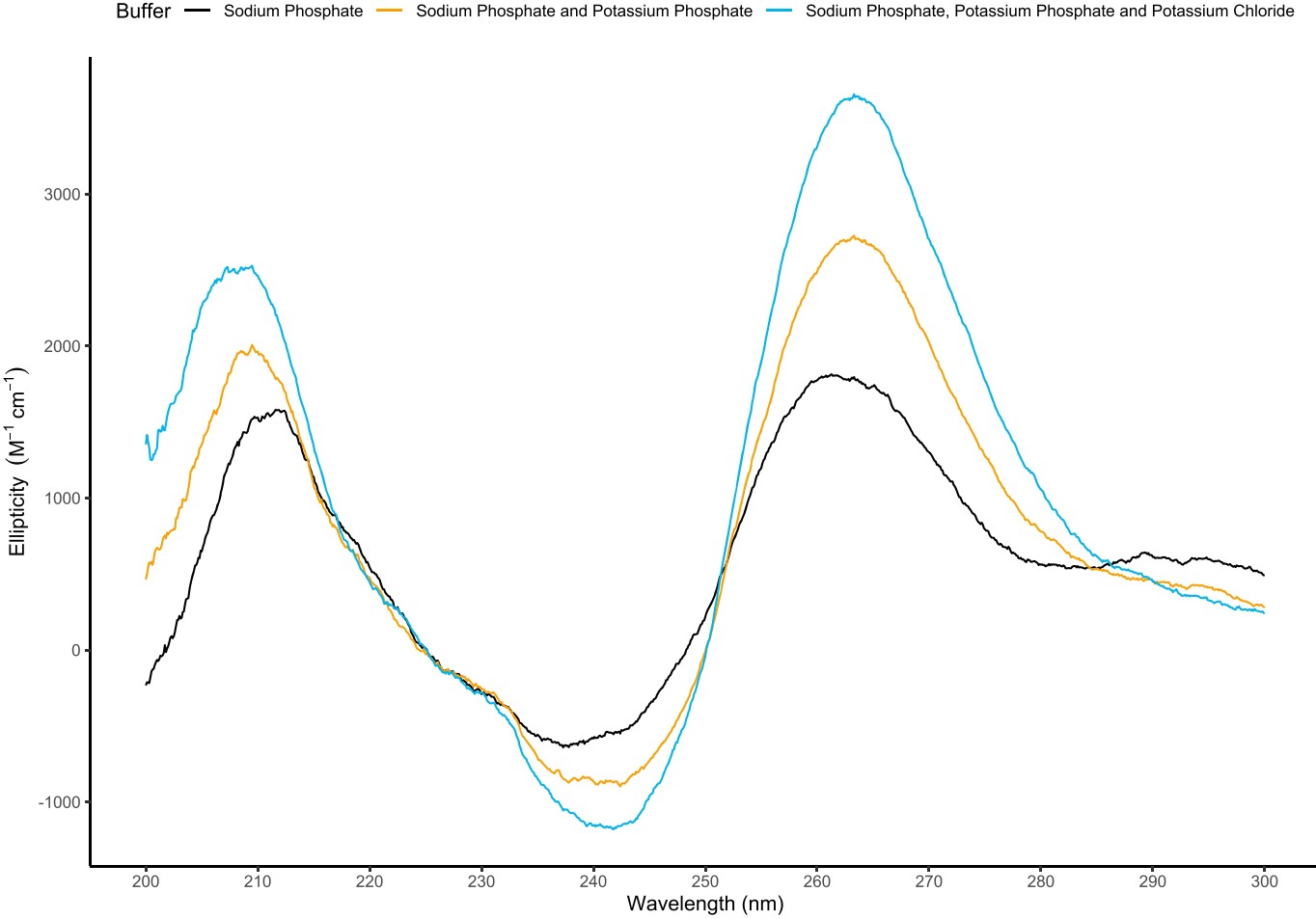

**Figure 6. Circular dichroism results of the TR21 consensus sequence oligonucleotide.**
The wavelength is shown on the x-axis with ellipticity on the y-axis. Three buffer solutions (pH 7.2) were used: 1 mM Na-phosphate (black), 1 mM Na-phosphate with 10 mM K-phosphate (orange), and 1 mM Na-phosphate with K-phosphate and K-chloride to 115 mM total K (blue). G4 signature peaks and troughs (210 and 260 nm peaks and a 240 nm trough) are displayed in all three buffers.

between *SLC6A3* gene expression in dopamine neurons as a function of repeat copy number at this site.

There are some limitations to this study. Although having 64 samples is considered small for association studies, the sample was sufficient to display the heterogeneous nature of each VNTR and to detail the VNTR landscape of *SLC6A3*. The costly nature of long-read sequencing entire genomes precludes analysis of larger numbers of chromosomes. In addition, although both CLR and CCS sequencing technologies were used to generate the long-read haplotype-phased genome assemblies, our comparison of sequencing technologies showed an average similarity of 99.96%, and there were no mismatches detected in any of the VNTR regions reported. Therefore, we used both technologies for our analysis because we were primarily focused on VNTR regions. In addition, manual annotation of TR identities across genome assemblies using output from the TRAL was required for detailed analysis (see the Materials and Methods section for details). Software that can perform this task automatically would minimize the potential for inadvertent errors. Finally, although the use of short-read data has been used as a proxy for analysis of long-read data (Kojima et al,

2016; Mukamel et al, 2021), the high GC content and high degree of variability in copy number of TR21 will require careful analysis to establish the reliability of using these methods. After validating these methods for TR21, association tests of genotypes of TR21 against multiple phenotypes using short-read data will be possible.

*SLC6A3* and its many polymorphisms have received considerable attention (Salatino-Oliveira et al, 2018). Association studies have linked many SNP and VNTR alleles to specific phenotypes of interest, but often with inconsistent results. Analysis of VNTRs within the gene using targeted long-read sequencing, in particular for TR21, a novel hyVNTR, might lead to strengthened and consistent associations with previously studied phenotypes.

## Materials and Methods

### Haplotype-phased long-read sequences of *SLC6A3*

We used 64 previously generated unrelated long-read haplotype-phased genome assemblies as our study sample (Ebert et al, 2021 –

version 1). Each assembly was available as a collection of contigs mapped to their respective chromosomes from which the contig that contained *SLC6A3* on chromosome 5 was extracted. To ensure that the entirety of *SLC6A3* was contained within each extracted contig, we searched for and confirmed the presence of two 30 bp sequences, located on either end of the gene (3′-UTR: CAGCG-GAAACGAGACAAGGAGGCTGAGGCAG (chr5:1,392,763–1,392,793) or its reverse complement and 5′-UTR: AGCCTCGGCCTCGGGCTCTTATC-CAGTAGA (chr5:1,445,441–1,445,470) or its reverse complement). After extracting the *SLC6A3* sequence from each contig, we oriented each gene in the 5′ to 3′ direction.

### Long-read sequencing technology comparison

Each CLR sequence was aligned to its corresponding CCS sequence using EMBOSS Kalign v3.3.1 (default parameters). Alignment identity matrices were used to determine the maximum number of mismatching base pairs, and pairwise alignment results were used to detect indels present between the two sequences.

### Tandem repeat annotation of *SLC6A3* sequences

TRs with unit length >6 bp were detected using PHOBOS (Mayer et al, 2010), TRF (Benson, 1999), T-REKS (Jorda & Kajava 2009), and XSTREAM (Newman & Cooper, 2007). Using TRALv2.0, a score was assigned to each TR using a phylogenetic model and divergence between repeat units was calculated. Repeats that were found to have arisen due to chance rather than duplication events (likelihood ratio test *P*-value > 0.05) and those with divergence > 0.05 were discarded. Using common-ancestry clustering, the remaining set of TRs was made nonredundant. In case of overlap between repeats, only the TR with the lowest *P*-value and divergence was retained. Finally, TRs were further refined by constructing cpHMMs for each repeat and re-annotating them in the original gene sequence for more sensitive detection (Schaper et al, 2014). (See https://github.com/maverbiest/dopamine_transporter_repeats for access to the code used to annotate tandem repeats).

Using the output from the TRAL, we manually assigned each annotated TR an ID if it was present in more than half of the *SLC6A3* sequences. Repeat ID consistency across samples was ensured by comparing repeat consensus sequences. After each TR was assigned an ID, we determined which repeats had more than one repeat copy number present across individuals in the sample. TRs with more than one allele present in our sample were considered VNTRs. In the case of TR17, the TRAL identified a few different length repeats including a 78 bp repeat that in analysis with TRF was found to be two copies of a 38 bp repeat, which was used in subsequent analyses to be more parsimonious.

### Comparing sequence identity of VNTRs with "Mola" charts

To compare both the individual repeat sequences within each VNTR, and these sequences across all *SLC6A3* assemblies, we generated what we have termed "Mola" charts, named after the famous Panamanian textile art. "Mola" charts graphically demonstrate similarities and differences between each individual repeat

sequence of the VNTRs. Two different types of Mola charts were generated, which we termed "multi-color" and "3-color."

For the multi-color Mola charts, every VNTR's unique sequence patterns were identified and assigned a different, random color. The unique repeats were given a color using the built in Visual Basic Editor in Microsoft Excel. Each individual's complete VNTR length was then plotted, using the assigned colors to represent the corresponding repeats. The resulting chart demonstrates both the diversity of sequence composition and unique repeat frequency for each VNTR. Plots with a high degree of variability in color would be considered to have a low-repeat sequence "purity," and plots with a low degree of variability in color would be considered to have a high-repeat sequence "purity."

Microsoft Excel was also used to generate 3-color Mola charts. First, a file with separate cells for each copy of a VNTR's repeat unit from each sample was created. This file was then used to generate a global alignment score for each repeat sequence with the VNTR's consensus sequence using an EMBOSS needle (Rice et al, 2000) in the Galaxy bioinformatics website (Jalili et al, 2020). A 3-color conditional formatting rule was then applied to the data so that each cell received a color, blue being the most closely aligned to the consensus, red as the least aligned, and white the median value.

### VNTR copy number inheritance patterns

Haplotype-phased long-read genome assemblies of three mother–father–child trios (HG00512, HG00513, and HG00514; HG00731, HG00732, and HG00733; and NA19238, NA19239, and NA19240) were also downloaded (Ebert et al, 2021). Each assembly was previously constructed by Ebert and colleagues using PacBio CCS or CLR reads. *SLC6A3* was extracted from each assembly and oriented as described above. In addition, all assemblies were annotated for TRs using TRALv2.0 as described above. The TR sites from the original sample of 64 sequences that were determined to be polymorphic in copy number were analyzed in each trio to determine patterns of inheritance. All VNTRs were present in both haplotypes of each child. After parent and child VNTR copy numbers were compared, each child's *SLC6A3* sequence was aligned to its corresponding parentally inherited sequence using the NCBI Global Align BLAST tool (found at https://blast.ncbi.nlm.nih.gov/Blast.cgi). To further test the fidelity of each VNTR during the inheritance process, each child's VNTR was aligned to the corresponding parental VNTR using an EMBOSS Needle v6.6.0 (default settings).

### Linkage disequilibrium and SNP–VNTR copy number correlations

All *SLC6A3* sequences were aligned to the human reference genome (GRCh38/hg38) using the MAFFTv7.504 web interface (added flags included "reorder," "keeplength," and "addfragments"). SNP-sites software (Page et al, 2016) was used to generate a variant calling file containing annotated SNPs and SNP allele frequencies for each *SLC6A3* sequence. LD values for each SNP–SNP, SNP–VNTR, and VNTR–VNTR pair were calculated by squaring their Pearson correlation coefficient using R statistical software (R v 4.1.2). An LD heatmap was produced using the *LDheatmap* function (Shin et al, 2006) in R (R v 4.1.2). All SNP–VNTR correlations were examined to determine if any previously

reported SNPs (dbSNPv153) were highly correlated with VNTR copy number. All VNTR–VNTR correlations were also examined to determine if any correlations between VNTR alleles were present. Correlations between all SNPs and VNTRs can be found in Table S6.

### G-quadruplex predictions and experimental data

G4-Hunter software was used to predict the presence of G4s in silico (http://bioinformatics.ibp.cz); "window" and "threshold" parameters were set to 25 and 1.6, respectively, as has been previously recommended (Bedrat et al, 2016). Potential areas of G4 formation were marked and compared with VNTR regions previously annotated. VNTRs that contained high G4 predictions values (greater than 50% of the region covered by G4s) were tested experimentally for the presence of G4s using CD spectroscopy. An oligonucleotide matching the consensus sequence of TR21 (GCCGGGGAGGGG-CAGGGCGGGGAAGGCGCTGGGTGGGGG) was purchased from Integrated DNA Technologies (Coralville, Iowa; https://www.idtdna.com/pages). The oligonucleotide was tested for G4 formation by taking spectra with a series of three buffer solutions (pH 7.2): first, 1 mM sodium phosphate (no potassium), then after addition of K-phosphate to a final concentration of 10 mM K, and finally after addition of K-chloride to a final K concentration of 115 mM (Kejnovská et al, 2019).

## Data Availability

All primary data used to generate haplotype assemblies are publicly available at INSDC under the following accessions and project IDs: Illumina high-coverage genomic sequence (PRJEB37677), HiC (ERP123231), Bionano Genomics (ERP124807), PacBio (PRJEB36100, ERP125611, and PRJNA698480), and Strand-seq (PRJEB39750). All haplotype assemblies used herein are publicly available at the following address: https://www.internationalgenome.org/data-portal/data-collection/hgsvc2.

## Supplementary Information

## Acknowledgements

This research was supported internal funds from the Pennsylvania State University and by the National Institute on Aging Grant T32 AG049676 to The Pennsylvania State University. We would also like to thank Larry John Stone for assistance with Excel data calculations.

### Author Contributions

AT Apsley: data curation, formal analysis, validation, investigation, visualization, methodology, and writing—original draft, review, and editing.

ER Domico: formal analysis, validation, investigation, visualization, methodology, and writing—review and editing.

MA Verbiest: software, formal analysis, validation, investigation, methodology, and writing—review and editing.

CA Brogan: formal analysis, validation, investigation, and writing—review and editing.

ER Buck: formal analysis, validation, investigation, visualization, methodology, and writing—review and editing.

AJ Burich: software, formal analysis, validation, investigation, methodology, and writing—review and editing.

KM Cardone: formal analysis, validation, investigation, visualization, methodology, and writing—review and editing.

WJ Stone: data curation, software, formal analysis, validation, investigation, methodology, and writing—review and editing.

M Anisimova: conceptualization, resources, software, formal analysis, supervision, funding acquisition, validation, project administration, and writing—review and editing.

DJ Vandenbergh: conceptualization, resources, formal analysis, supervision, funding acquisition, validation, investigation, visualization, methodology, project administration, and writing—original draft, review, and editing.

### Conflict of Interest Statement

The authors declare that they have no conflict of interest.

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
