## [Reviewer comments · Life Science Alliance]

Life Science Alliance

A Novel Hyper-Variable Variable Number Tandem Repeat in the Dopamine Transporter Gene (SLC6A3)

Abner Apsley, Emma Domico, Max Verbiest, Carly Brogan, Evan Buck, Andrew Burich, Kathleen Cardone, Wesley Stone, Maria Anisimova, and David Vandenberg

DOI: <https://doi.org/10.26508/lsa.202201677>

Corresponding author(s): David Vandenberg, Pennsylvania State University and Abner Apsley, Pennsylvania State University

Review Timeline:

Submission Date:	2022-08-17
Editorial Decision:	2022-10-13
Revision Received:	2022-12-15
Editorial Decision:	2023-01-10
Revision Received:	2023-01-25
Accepted:	2023-01-26

Transaction Report:

October 13, 2022

Re: Life Science Alliance manuscript #LSA-2022-01677-T

David J Vandenberg
Pennsylvania State University
Biobehavioral Health
258A Health & Hum Dev Bld
University Park, PA 16802

Dear Dr. Vandenberg,

Thank you for submitting your manuscript entitled "Characterization of a Novel Hyper-Variable Variable Number Tandem Repeat in the Dopamine Transporter Gene (SLC6A3)" to Life Science Alliance. The manuscript was assessed by expert reviewers, whose comments are appended to this letter. We invite you to submit a revised manuscript addressing the Reviewer comments.

Thank you for this interesting contribution to Life Science Alliance. We are looking forward to receiving your revised manuscript.

Sincerely,

B. MANUSCRIPT ORGANIZATION AND FORMATTING:

Reviewer #1 (Comments to the Authors (Required)):

Apsley and colleagues detail features of VNTRs in SLC6A3, which have been implicated in several disorders. They identify a novel hyper-variable VNTR in intron 8 with 3-133 copies in humans and leverage phased long-read sequence information to deduce the exact repeat sequence and internal nucleotide sequence composition. Further analysis of the TR21 repeat revealed presence of multiple predicted PRDM9 binding sites, along with a G-quadruplex structure that was confirmed by CD studies. This work is one of the first to perform detailed analysis of the complete sequence of multiple tandem repeats in one gene in multiple genomes and sets the stage for further analysis of the TR21 VNTR in disease.

- One consideration for TR21 is that the repeat itself is very GC-rich as seen in supplementary table 3 with long stretches of cytosines (or guanines depending on the orientation) and long-read sequencing approaches that are used are still imprecise at assigning the correct number of nucleotides at these homopolymer stretches. Therefore, the TR21 repeat motif composition may not be as diverse as it appears. If homopolymer stretch variation is indeed reflective of sequence inconsistency and not true genetic changes, some of the repeat motifs could be collapsed together and this could help reveal a more obvious pattern of expansion at this VNTR across genomes. In other words, it would be interesting to note whether any of the colored bands for TR21 are similar if differences in the C-rich stretches are considered only sequence errors. Comparing between CLR and CCS sequencing methods for the same individual could help provide an indication of the extent of this variation, and the 1-5 nucleotide differences in parent-child trios may also simply be sequence errors if they arise in homopolymer stretches. In any event, this sequencing limitation should at least be mentioned.

- One limitation of the study is that it does not assess any connection to disease state despite the many associations mentioned for other VNTRs in SLC6A3. The abstract seems to suggest that the authors found VNTRs that account for the genetic heterogeneity. Many of the samples with long-read sequencing data also have high coverage short read data available, which can be used to estimate repeat length based on the density of reads at the repeat relative to baseline genome coverage. Is there a correlation between long-read and short read data for these individuals? If so, the information can be used to guide researchers that have access to short read WGS data to estimate repeat length differences in various disease states.

- Suggesting that the SLC6A3 is a recombination hotspot in the abstract is a bit speculative since only correlative data has been obtained and not functional validation; consider changing to "may be" a recombination hotspot.

- Is there any indication that the repeat length of TR21 or the subgroups of TR17 cluster by 1000 Genomes project superpopulation?

- Scale bars should be provided for figure 2. As displayed, it appears that a similar amount of deviation from the reference repeat motif is present for each VNTR, but I suspect that this is not the case and that TR21 has much more variability in sequence composition. Providing the information on a scale with the number of nucleotide differences (or indels) between the most abundant repeat motif and each of the subsequent motifs in the VNTR might help the reader appreciate the amount of variability that is taking place and where within the repeat itself this is most notable.

- Buffer concentrations used for figure 5 should be explicitly stated in the methods in addition to referencing the protocol that was used.

- The authors mention there is ambiguity about the length of the TR21 motif length (38 or 75 bp). Based on long-read assessment, if there was a bias for instance to even numbers of motifs (4, 6, 8, etc.), it would suggest that the effective repeat motif is actually the longer one. If there is an equal distribution of even and odd repeat lengths, then the 38bp motif is more likely the correct length.

Minor points

- The source for the phased genomes from Ebert et al should be provided (i.e. the Human Genome Structural Variation Consortium (HGSVC)), and the reference should be updated to reflect the final manuscript and not a psu.edu weblink
- "Indels" shouldn't be in all caps.

- Recommend adding %GC for each VNTR in table 2

Reviewer #2 (Comments to the Authors (Required)):

Apsley and colleagues present a focused study of the biomedically important SLC6A3 locus. The authors use a recently published dataset comprising phased whole-genome assemblies of 32 samples (64 haplotypes) to examine the VNTR composition of the locus in detail. The presented analyses include assessments of the observed variation, genomic stability and LD taking other variants in the locus into account. A central hypothesis in the manuscript is that contradictory studies on SLC6A3 so far published could be explained by incomplete or insufficient information about the VNTR composition of SLC6A3. The authors' contribution certainly improves the characterization of the SLC6A3 locus, although some question marks regarding overall novelty and applied procedures remain in the current version of the manuscript.

Major:

P5/6: please add a pointer to the supplement and list the various VNTR consensus sequences you are referring to here.

P6: "inconsistent and contradicting results" - can you add a simple example here such that non-experts for SLC6A3 can get an impression what type of inconsistencies/contradictions have been published?

P6: what does "version 1" in the reference to the Ebert et al. paper indicate?

P9: failed PCR: can you explain, i.e., formulate a plausible hypothesis, why the PCR failed? Please add a brief summary to the supplement to make this more comprehensible

P10-12: I believe something is off about the figure numbering

P11: though I can somehow distinguish between "high and low variability", I am not sure what I should see in these Mola charts?

It's absolutely impossible to read the labels; at the very least, you should consider adding a color code for the five continental groups of the assemblies/samples. I would assume the observed variation clusters by these groups?

P11 fig. 1 legend: "unit length is from TRAL, except for TR17" - why this exception?

P13: how did you "observe" the SNPs?

P13: how did you estimate/derive the allele frequencies?

P16/17: I find your assessment of TR21 quite confusing; is it really novel, or just - due to technological progress - a more specific characterization of a VNTR already known (Byerley et al.)?

P17: I am unclear about your tentative statement regarding PRDM9 motif disruptions. I understand that you annotated the sequences with the respective motif, and found quite a lot (421 for the largest sample). So is there any evidence for disrupted PRDM9 motifs?

P18/19: "manual annotation of TR identities...[and so on]" - ok, true, but what do you want the reader to make with that information? Additionally, and more importantly, the results of this manual annotation process of yours should thus be made publicly available for others to use and check (I don't see anything related to that in the Data availability statement).

P24: I am missing a statement (link) about the public accessibility for the code you used for your study (running TR detection algorithms, TRAL etc.)

Minor:

P4: "was also tested in many other studies", but you cite only one?

P5: please state the coordinates of SLC6A3 on chr5 (hg38)

P8: your section about TRAL could use a pointer to the supplement/Methods to make clear that you used several TR detection algorithms (you just mention TRF here)

P11: "similar similarities" seems quite redundant

P12: "...and their parents in the original sample" - what original sample are you talking about here?

P18: "we felt justified in the use of both technologies" - that sounds a bit odd in its defensiveness.

P21: What are Excel developer tools? Reference?

Discretionary:

P13: "...a Region of Very Low Linkage..." - style: "very" is often regarded as bad style, because it's overly subjective in the sense that it's unclear where you draw the line between "low" and "very low".

We greatly appreciate the time and effort you spent to review this manuscript. We have enumerated all of the reviewer's concerns in bold below and listed our responses under each concern. We also provided the reviewers summary as a refresher.

Reviewer #1 Summary:

Apsley and colleagues detail features of VNTRs in SLC6A3, which have been implicated in several disorders. They identify a novel hyper-variable VNTR in intron 8 with 3-133 copies in humans and leverage phased long-read sequence information to deduce the exact repeat sequence and internal nucleotide sequence composition. Further analysis of the TR21 repeat revealed presence of multiple predicted PRDM9 binding sites, along with a G-quadruplex structure that was confirmed by CD studies. This work is one of the first to perform detailed analysis of the complete sequence of multiple tandem repeats in one gene in multiple genomes and sets the stage for further analysis of the TR21 VNTR in disease.

Reviewer #1 Concerns: Major comments (1-7) and Minor comments (8-11)

- 1. A) One consideration for TR21 is that the repeat itself is very GC-rich as seen in supplementary table 3 with long stretches of cytosines (or guanines depending on the orientation) and long-read sequencing approaches that are used are still imprecise at assigning the correct number of nucleotides at these homopolymer stretches. Therefore, the TR21 repeat motif composition may not be as diverse as it appears. If homopolymer stretch variation is indeed reflective of sequence inconsistency and not true genetic changes, some of the repeat motifs could be collapsed together and this could help reveal a more obvious pattern of expansion at this VNTR across genomes. In other words, it would be interesting to note whether any of the colored bands for TR21 are similar if differences if the C-rich stretches are considered only sequence errors.**

Thank you for this comment. We have performed an additional analysis in which we have omitted the homopolymer runs from all TR21 unique sequences. We performed this analysis in a step-by-step process where we first excluded homopolymer runs of 6 nucleotides, then of 6 and 4 nucleotides (there were no runs of 5), and finally of 6, 4, and 3 nucleotides. With all homopolymer runs included, there were 206 unique repeat sequences. When we excluded 6 nucleotide homopolymer runs, there were still 206 unique repeat sequences. When we excluded 6 and 4 homopolymer runs, we observed 200 unique repeat sequences. And when we excluded homopolymer runs of 6, 4, and 3, we observed 179 unique repeat sequences. This

lack of a large change in unique sequences suggests that the variation observed in TR21 did not arise due to sequencing errors, but we cannot conclude definitively one way or the other. The next paragraphs include additional analyses we have conducted.

To represent these results, we included the following in our Results section: “To ensure that the large variation observed in TR21 was not due to homopolymer run sequencing errors we performed a sensitivity analysis by excluding homopolymer runs in repeat sequences of TR21. When excluding homopolymer runs of 4 nucleotides or longer, only six of the unique sequences were the same as others, leaving 200 unique TR21 repeat sequences. This minimal change in the number of unique repeat sequences after excluding homopolymer runs suggests that the variation observed in TR21 was not due to sequencing errors. Additional analysis of the same chromosome in parent-child pairs generated similar results (see our response to part B of this concern).”

We have seen, however, that when there are differences between repeats from parent-child comparisons, the variation tends to be present at the end of a string of Cytosines. This result still does not clarify if the difference is due to a sequencing error, or an actual polymorphism. We have also added the following three paragraphs, along with an additional paragraph from part B of this concern, to the supplemental information (in the file named Supplemental_Information):

“The CCS group was calculated to have an average of 1.833 ± 0.749 gaps between parent child pairs. The mean value for the number of gaps between parent child pairs in the CLR group was 17 ± 5.235 . While only 6 total parent child pairs were used in this comparison, there was a clear trend since the CLR group had more gaps than the CCS in every pairing while also having nearly 10 times the total number of gaps. There were also no mismatches seen in any of the CCS pairings while one of the parent-child pairs in the CLR group had 2 mismatches. This lower alignment seen in the CLR group implies that it is not as accurate of a sequencing method as CCS, which is further supported by CCS being considered high fidelity.

While it is possible that the CCS sequencing is misregistering actual polymorphisms that CLR sequencing didn't, it is very unlikely that it would cut these polymorphisms in such a stable manner as to significantly increase the alignment of these sequences which would have to be the case unless it is more accurate than CLR.

Almost all of the gaps were seen in long homopolymer C stretches followed by an A. These homopolymer stretches are known to be difficult to sequence accurately, so it is reasonable that these gaps are misreads. However, these homopolymer stretches also have a high rate of polymorphism due to replication slippage causing actual misreads of the DNA by the replication machinery. So, we are unable to consider all gaps seen as just misreads, and we do not have the data to resolve this issue.”

B) Comparing between CLR and CCS sequencing methods for the same individual could help provide an indication of the extent of this variation, and the 1-5 nucleotide differences in parent-child trios may also simply be sequence errors if they arise in homopolymer stretches. In any event, this sequencing limitation should at least be mentioned.

This is a great suggestion. We have added the following text to the supplemental file along with the text from the comment above:

“We compared CLR and CCS sequencing results from TR21 in 12 sequences. These 12 sequences showed a variation in the presence of discrepancies. The maximum number of mismatches was 2bp, with most comparisons having no mismatches. The maximum number of indels was 39, with 34 of the errors being in a homopolymer stretch. The mean number of indels was 11.9 and most of these indels were in homopolymer runs. Although these errors were present, according to our results described in concern 1A, we believe that these errors were not significantly influential on the number of unique sequences in TR21.”

2. **A) One limitation of the study is that it does not assess any connection to disease state despite the many associations mentioned for other VNTRs in SLC6A3. The abstract seems to suggest that the authors found VNTRs that account for the genetic heterogeneity.**

We agree and have changed the wording of the Abstract to the following: “We searched for unanalyzed VNTRs in SLC6A3, *to enable future work to test for consistent* association study results.”

B) Many of the samples with long-read sequencing data also have high coverage short read data available, which can be used to estimate repeat length based on the density of reads at the repeat relative to baseline genome coverage. Is there a correlation between long-read and short read data for these individuals? If so, the information can be used to guide researchers that have access to short read WGS data to estimate repeat length differences in various disease states.

This work is our next step, and while the use of short-read data has been used as a proxy for analysis of long-read data, the high GC content and high degree of variability in repeat number of this hyperVNTR will require careful analysis to establish the reliability of short reads, and then we can test genotypes of the hyperVNTR against multiple phenotypes. Given these details, we feel the work is beyond the scope of this paper. We have, however, included a citation to Kojima et al., 2016, *BMC Genomics*, and Mukamel et al., 2021, *Science*, that detail algorithms to estimate tandem repeat copy number using short-read WGS data. We have incorporated this citation into the Discussion section of our article to point researchers in the right direction if they only have short-read sequencing data.

3. **Suggesting that the SLC6A3 is a recombination hotspot in the abstract is a bit speculative since only correlative data has been obtained and not functional validation; consider changing to "may be" a recombination hotspot.**

Thank you for this suggestion. We have changed the wording in the Abstract to more accurately reflect that this VNTR is only potentially a recombination hotspot.

4. **Is there any indication that the repeat length of TR21 or the subgroups of TR17 cluster by 1000 Genomes project superpopulation?**

Great question! We tested for this but did not put the results in the manuscript. We have added a sentence in the Results mentioning that this VNTR does not cluster by 1000 Genomes project's superpopulations or ancestry. We also included an additional supplemental table (Table S4) to show these results with all other VNTRs analyzed. Our new paragraph in the Discussion reads as follows: "In addition to constructing multi-colored and three-colored Mola charts, we also tested whether alleles to each VNTR could be grouped based on ancestry or 1000 Genomes Project superpopulation. Ancestry groups included African (haplotype sample size n=16), South Asian (n=10), East Asian (n=12), European (n=12), African admixed (n=2), and Indigenous admixed (n=10). After excluding the African admixed ancestry category due to a low sample size, ancestry group did not predict TR21 copy number ($F(4,55)=1.77$, $p=0.15$). Results for TR21 alleles grouped by ancestry can be found in Table S4."

- 5. Scale bars should be provided for figure 2. As displayed, it appears that a similar amount of deviation from the reference repeat motif is present for each VNTR, but I suspect that this is not the case and that TR21 has much more variability in sequence composition. Providing the information on a scale with the number of nucleotide differences (or indels) between the most abundant repeat motif and each of the subsequent motifs in the VNTR might help the reader appreciate the amount of variability that is taking place and where within the repeat itself this is most notable.**

This is a great comment! We have added the highest % nucleotide variation of each repeat to the Results section. We have also more clearly explained that the color patterns in Figure 3 show the VNTR's *relative* amount of variation from the repeat's consensus sequence. Results detailing Figure 3 now read: "Three-colored Mola charts for each VNTR are shown in Figure 3. The colors are based on sequence similarity to the individual VNTR's consensus (global alignment score, EMBOSS needle, Rice et al., 2000), which emphasizes the relative degree of variation from the consensus present in each VNTR. The greatest differences in repeat sequences from the consensus sequence, for each VNTR, are as follows: TR09 gaps/mismatches = 26, TR17 gaps/mismatches = 27, TR21 gaps/mismatches = 15, TR22 gaps/mismatches = 3, TR30 gaps/mismatches = 5."

- 6. Buffer concentrations used for figure 5 should be explicitly stated in the methods in addition to referencing the protocol that was used.**

We agree and have added the concentrations of buffers used in the methods, and more succinctly in the legend for Figure 2.

- 7. The authors mention there is ambiguity about the length of the TR17 motif length (38 or 75 bp). Based on long-read assessment, if there was a bias for instance to even numbers of motifs (4, 6, 8, etc.), it would suggest that the effective repeat motif is actually the longer one. If there is an equal distribution of even and odd repeat lengths, then the 38bp motif is more likely the correct length.**

Thank you for this suggestion. We calculated the percentage of alleles that were even and odd for TR17 to determine if 38 or 75 was the more accurate consensus sequence length. 28% (18/64) of repeat alleles were odd numbered and 72% (46/64) were even numbered; however,

we decided to maintain the consensus sequence length of TR17 at 38 because the second half of the 75 bp consensus sequence is almost identical to the first half and a smaller repeat would be more parsimonious. We have added this description to the last paragraph in the Methods section titled, "Tandem Repeat Annotation of *SLC6A3* Sequences."

8. The source for the phased genomes from Ebert et al should be provided (i.e., the Human Genome Structural Variation Consortium (HGSVC)).

Thank you for this suggestion. We have updated the first reference to these sequences in paragraph 5 of the Introduction section. It now reads "In the present work, we used 64 publicly available long-read haplotype-phased genome assemblies from 32 individuals from the Human Genome Structural Variation Consortium (HGSVC)...".

9. The reference to Ebert should be updated to reflect the final manuscript and not a psu.edu weblink

Thank you for pointing out this error, we have updated the reference to reflect the final manuscript.

10. "Indels" shouldn't be in all caps.

We have corrected all instances where we used the phrase "INDELS" to "Indels" or "indels".

11. Recommend adding %GC for each VNTR in table 2

We have added %GC for each VNTR in Table 2.

Reviewer #2 Summary:

Apsley and colleagues present a focused study of the biomedically important *SLC6A3* locus. The authors use a recently published dataset comprising phased whole-genome assemblies of 32 samples (64 haplotypes) to examine the VNTR composition of the locus in detail. The presented analyses include assessments of the observed variation, genomic stability and LD taking other variants in the locus into account. A central hypothesis in the manuscript is that contradictory studies on *SLC6A3* so far published could be explained by incomplete or insufficient information about the VNTR composition of *SLC6A3*. The authors' contribution certainly improves the characterization of the *SLC6A3* locus, although some question marks regarding overall novelty and applied procedures remain in the current version of the manuscript.

Reviewer #2 Concerns: Major comments (12-24) and Minor comments (25-32)

12. P5/6: please add a pointer to the supplement and list the various VNTR consensus sequences you are referring to here.

Thank you for this suggestion. We have added pointers in these paragraphs to the VNTR consensus sequences and other information listed in Tables 1-2.

13.P6: "inconsistent and contradicting results" - can you add a simple example here such that non-experts for SLC6A3 can get an impression what type of inconsistencies/contradictions have been published?

Good suggestion; we have added the following sentence in the paragraph of interest: "For example, in meta-analyses and systematic reviews, alcohol addiction has been reported to be both significantly and not significantly associated with the previously described 3'-UTR VNTR genotype (van der Zwaluw et al. 2009; Ma, Fan, and Li, 2016)".

14.P6: what does "version 1" in the reference to the Ebert et al. paper indicate?

We agree that this reference to a version is confusing and we have dropped it. We had included that because we have had communication with Dr. Ebert indicating that there is effort to generate more accurate sequence. We realize that any new version of the sequence will be annotated and possibly associated with a different publication so it is best to limit the description to a simple statement.

15.P9: failed PCR: can you explain, i.e., formulate a plausible hypothesis, why the PCR failed? Please add a brief summary to the supplement to make this more comprehensible.

We have modified the sentence that raises the issue to say, "This site was not amplified by PCR despite numerous attempts using more than 7 different commercial DNA polymerase kits." This sentence is followed by a sentence mentioning high G+C content and G-Quadruplex formation as possible reasons for the inability to generate DNA by PCR. We considered adding the list of enzymes tried as a supplement, and we can, if the reviewers desire. The list would include the following enzymes, but we are not sure it would be of value:

AmpliTaq Gold (Thermo Fisher), Platinum SuperFi (Invitrogen), HS VeriFi Mix (PCR Biosystems), Q5 (New England Biolabs), Long and Accurate (TaKaRa), Velocity (BioLine), and PrimeStar (also TaKaRa). Basically, the corresponding author searched for new commercial enzymes at genetics and neuroscience meetings over the past 20 years and the lab tried them all without success.

16.P10-12: I believe something is off about the figure numbering

Thank you for pointing this out. We have reviewed the figure numbering and believe everything is correct but are happy to make any changes if you notice any further issues.

17.P11: though I can somehow distinguish between "high and low variability", I am not sure what I should see in these Mola charts? It's absolutely impossible to read the labels; at the very least, you should consider adding a color code for the five continental groups of the assemblies/samples. I would assume the observed variation clusters by these groups?

Thank you for these suggestions. We have removed the labels for individual haplotypes so as to not confuse the reader. We have also included an additional supplementary table (Table S4),

that groups alleles for each VNTR by ancestry. Additionally, we have addressed the ancestry clustering question in concern 4 of this document.

18.P11 fig. 1 legend: "unit length is from TRAL, except for TR17" - why this exception?

Thank you for pointing this out. Our pipeline of annotating TRs in this gene has more clearly been spelled out in the Methods section "Tandem Repeat Annotation of SLC6A3 Sequences", which discusses why TR17 was an exception. Concern 23 in this document also briefly touches on this issue. Due to the overlap in concerns, we have responded to concern 23 and have excluded this sentence ("unit length is from TRAL, except for TR17") from the figure legend.

19.P13: how did you "observe" the SNPs?

We have added a comment in this paragraph directing the reader to the Methods section. Here, we detail our process in annotating SNPs.

20.P13: how did you estimate/derive the allele frequencies?

Similar to concern 19, we have added a comment in this paragraph directing the reader to the Methods section. Here, we detail our process in determining SNP allele frequencies.

21.P16/17: I find your assessment of TR21 quite confusing; is it really novel, or just - due to technological progress - a more specific characterization of a VNTR already known (Byerley et al.)?

This important point has been expanded on (p 18) to clarify that Byerley's data probably detected variation at 3 tandem repeats, but the possibility still exists that the extra length of alleles from Byerley could be due to variability at TR21.

22.P17: I am unclear about your tentative statement regarding PRDM9 motif disruptions. I understand that you annotated the sequences with the respective motif, and found quite a lot (421 for the largest sample). So is there any evidence for disrupted PRDM9 motifs?

Thank you for voicing this concern. We have changed the wording in the Discussion to the following: "Although not tested in the present work, disruptions of PRDM9 binding site motifs in TR21 may take place as a result of the impurity of this hyVNTR's repeat sequences". We hope this wording conveys a more accurate representation of the results we actually present and is not too speculative.

23.P18/19: "manual annotation of TR identities...[and so on]" - ok, true, but what do you want the reader to make with that information? Additionally, and more importantly, the results of this manual annotation process of yours should thus be made publicly available for others to use and check (I don't see anything related to that in the Data availability statement).

Thank you for this comment. We have included a github link to the TRAL code that was used to annotate the sequences (see comment 24 as well). Additionally, we have detailed the manual annotation process of repeat IDs in the Methods section more clearly. It now reads as follows: "Using the output from TRAL, we manually assigned each annotated TR an ID if it was present in more than half of the SLC6A3 sequences. Repeat ID consistency across samples was ensured by comparing repeat consensus sequences."

24.P24: I am missing a statement (link) about the public accessibility for the code you used for your study (running TR detection algorithms, TRAL etc.)

We have included a link to our GitHub page where we provided code that was used for tandem repeat annotations in this study. This link was referenced in our Methods section as follows: "(See https://github.com/maverbiest/dopamine_transporter_repeats for access to the code used to annotate tandem repeats)."

25.P4: "was also tested in many other studies", but you cite only one?

Thank you for pointing this out. In our citation, we have noted that this citation is an extensive review and has many examples detailed therein. The wording has been changed to: "This intron-8 VNTR was also tested in many other studies for its association with disease-related phenotypes (see Salatino-Oliveira et al. 2018 for an extensive review).

26.P5: please state the coordinates of SLC6A3 on chr5 (hg38)

We have added the coordinates of SLC6A3 to paragraph 3 of the Introduction.

27.P8: your section about TRAL could use a pointer to the supplement/Methods to make clear that you used several TR detection algorithms (you just mention TRF here)

Thank you for this comment. We have added a pointer to the Methods section here as well as listed the different algorithms that were used in addition to TRF.

28.P11: "similar similarities" seems quite redundant

It might even be considered repetitively redundant! We have changed the wording of this sentence to read as follows: "Similarities in both length and number of units...".

29.P12: "...and their parents in the original sample" - what original sample are you talking about here?

We have included a more detailed description of the sequences that we used here that does not reference a "sample", but instead references the sequences we were using for all other analyses, and 6 additional sequences that were available from children of couples from our original sequences.

30. P18: "we felt justified in the use of both technologies" - that sounds a bit odd in its defensiveness.

Thank you for pointing this out. We have changed the wording of this sentence to read: "we used both technologies...".

31. P21: What are Excel developer tools? Reference?

The Excel tools used were built-in basic tools in Excel, so we have reworded this sentence to be more accurate. The section in the manuscript that references these tools now reads as follows: "The unique repeats were given a color using the built in Visual Basic Editor in Microsoft Excel..."

32. P13: "...a Region of Very Low Linkage..." - style: "very" is often regarded as bad style, because it's overly subjective in the sense that it's unclear where you draw the line between "low" and "very low".

We have edited the section heading in the Results to not include the word "very."

In sum, we hope that we have fully addressed the reviewers' concerns. We felt that the changes improved the manuscript greatly and we appreciate their efforts. Please don't hesitate to contact us if anything needs further clarification.

January 10, 2023

RE: Life Science Alliance Manuscript #LSA-2022-01677-TR

Dr. David J Vandenberg
Pennsylvania State University
Biobehavioral Health
258A Health & Hum Dev Bld
University Park, PA 16802

Dear Dr. Vandenberg,

Thank you for submitting your revised manuscript entitled "A Novel Hyper-Variable Variable Number Tandem Repeat in the Dopamine Transporter Gene (SLC6A3)". We would be happy to publish your paper in Life Science Alliance pending final revisions necessary to meet our formatting guidelines.

- please add the author contributions to the main manuscript text
- thank you for providing the ORCID ID's for the authors in the manuscript file, however, due to privacy laws, every author has to connect their ORCID ID to their LSA account on their own; please follow the instructions provided in the emails from LSA and get in touch with us at contact@life-science-alliance.org if you have any trouble
- please add your table legends to the main manuscript text
- the Supplemental Material file can be incorporated into the main manuscript text

A. FINAL FILES:

B. MANUSCRIPT ORGANIZATION AND FORMATTING:

**Submission of a paper that does not conform to Life Science Alliance guidelines will delay the acceptance of your

manuscript.**

The license to publish form must be signed before your manuscript can be sent to production. A link to the electronic license to publish form will be sent to the corresponding author only. Please take a moment to check your funder requirements.

Sincerely,

Reviewer #1 (Comments to the Authors (Required)):

The authors have made satisfactory revisions to the manuscript and addressed all of my concerns.

Reviewer #2 (Comments to the Authors (Required)):

The authors have adequately addressed all my concerns and I have no further comments.

January 26, 2023

RE: Life Science Alliance Manuscript #LSA-2022-01677-TRR

Dr. David J Vandenberg
Pennsylvania State University
Biobehavioral Health
258A Health & Hum Dev Bld
University Park, PA 16802

Dear Dr. Vandenberg,

Thank you for submitting your Research Article entitled "A Novel Hyper-Variable Variable Number Tandem Repeat in the Dopamine Transporter Gene (SLC6A3)". It is a pleasure to let you know that your manuscript is now accepted for publication in Life Science Alliance. Congratulations on this interesting work.

DISTRIBUTION OF MATERIALS:

Again, congratulations on a very nice paper. I hope you found the review process to be constructive and are pleased with how the manuscript was handled editorially. We look forward to future exciting submissions from your lab.

Sincerely,
